# cpSGD: Communication-efficient and differentially-private distributed SGD

**Naman Agarwal**
Google Brain
Princeton, NJ 08540
namanagarwal@google.com

**Ananda Theertha Suresh**
Google Research
New York, NY
theertha@google.com

**Felix Yu**
Google Research
New York, NY
felixyu@google.com

**Sanjiv Kumar**
Google Research
New York, NY
sanjivk@google.com

**H. Brendan McMahan**
Google Research
Seattle, WA
mcmahan@google.com

## Abstract

Distributed stochastic gradient descent is an important subroutine in distributed learning. A setting of particular interest is when the clients are mobile devices, where two important concerns are communication efficiency and the privacy of the clients. Several recent works have focused on reducing the communication cost or introducing privacy guarantees, but none of the proposed communication efficient methods are known to be privacy preserving and none of the known privacy mechanisms are known to be communication efficient. To this end, we study algorithms that achieve both communication efficiency and differential privacy. For $d$ variables and $n \approx d$ clients, the proposed method uses $\mathcal{O}(\log \log(nd))$ bits of communication per client per coordinate and ensures constant privacy.

We also improve previous analysis of the *Binomial mechanism* showing that it achieves nearly the same utility as the Gaussian mechanism, while requiring fewer representation bits, which can be of independent interest.

## 1 Introduction

### 1.1 Background

Distributed stochastic gradient descent (SGD) is a basic building block of modern machine learning [25, 11, 9, 28, 1, 27, 5]. In the typical scenario of synchronous distributed learning, in every round, each client obtains a copy of a global model which it updates based on its local data. The updates (usually in the form of gradients) are sent to a parameter server, where they are averaged and used to update the global model. Alternatively, without a central server, each client maintains a global model and either broadcasts the gradient to all or a subset of other clients, and updates its model with the aggregated gradient. In our paper we specifically consider the centralized setting, for the decentralized case the authors are referred to [36] and references therein.

Often, the communication cost of sending the gradient becomes the bottleneck [30, 23, 22]. To address this issue, several recent works have focused on reducing the communication cost of distributed learning algorithms via gradient quantization and sparsification [32, 17, 33, 20, 21, 4, 34]. These algorithms have been shown to improve communication cost and hence communication time in distributed learning. This is especially effective in the federated learning setting where clients are mobile devices with expensive up-link communication cost [26, 20].

While communication is a key concern in client based distributed machine learning, an equally important consideration is that of protecting the privacy of participating clients and their sensitive information. Providing rigorous privacy guarantees for machine learning applications has been an area of active recent interest [6, 35, 31]. Differentially private gradient descent algorithms in particular were studied in the work of [2]. A direct application of these mechanisms in distributed settings leads to algorithms with high communication costs. The key focus of our paper is to analyze mechanisms that achieve rigorous privacy guarantees as well as have communication efficiency.

## 1.2 Communication efficiency

We first describe synchronous distributed SGD formally. Let $F(w) : \mathbb{R}^d \to \mathbb{R}$ be of the form $F(w) = \frac{1}{M} \cdot \sum_{i=1}^{M} f_i(w)$, where each $f_i$ resides at the $i^{th}$ client. For example, $w$'s are weights of a neural network and $f_i(w)$ is the loss of the network on data located on client $i$. Let $w^0$ be the initial value. At round $t$, the server transmits $w^t$ to all the clients and asks a random set of $n$ (batch size / lot size) clients to transmit their local gradient estimates $g_i^t(w^t)$. Let $S$ be the subset of clients. The server updates as follows

$$g^t(w^t) = \frac{1}{n} \sum_{i \in S} g_i^t(w^t), \qquad w^{t+1} \triangleq w^t - \gamma g^t(w^t)$$

for some suitable choice of $\gamma$. Other optimization algorithms such as momentum, Adagrad, or Adam can also be used instead of the SGD step above.

Naively for the above protocol, each of the $n$ clients needs to transmit $d$ reals, typically using $\mathcal{O}(d \cdot \log 1/\eta)$ bits[1]. This communication cost can be prohibitive, e.g., for a medium size PennTreeBank language model [39], the number of parameters $d > 10$ million and hence total cost is $\sim 38$MB (assuming 32 bit float), which is too large to be sent from a mobile phone to the server at every round.

Motivated by the need for communication efficient protocols, various quantization algorithms have been proposed to reduce the communication cost [33, 20, 21, 38, 37, 34, 5]. In these protocols, the clients quantize the gradient by a function $q$ and send an efficient representation of $q(g_i^t(w^t))$ instead of its actual local gradient $g_i^t(w^t)$. The server computes the gradient as

$$\tilde{g}^t(w^t) = \frac{1}{n} \sum_{i \in S} q(g_i^t(w^t)),$$

and updates $w^t$ as before. Specifically, [33] proposes a quantization algorithm which reduces the requirement of full (or floating point) arithmetic precision to a bit or few bits per value on average. There are many subsequent works e.g., see [21] and in particular [5] showed that stochastic quantization and Elias coding [15] can be used to obtain communication-optimal SGD for convex functions. If the expected communication cost at every round $t$ is bounded by $c$, then the total communication cost of the modified gradient descent is at most

$$T \cdot c. \tag{1}$$

All the previous papers relate the error in gradient compression to SGD convergence. We first state one such result for completeness for non-convex functions and prove it in Appendix A. Similar (and stronger) results can be obtained for (strongly) convex functions using results in [16] and [29].

**Corollary 1** ([16]). *Let $F$ be $L$-smooth and $\forall x \, \|\nabla F(x)\|_2 \leq D$. Let $w^0$ satisfy $F(w^0) - F(w^*) \leq D_F$. Let $q$ be a quantization scheme, and $\gamma \triangleq \min\left\{L^{-1}, \sqrt{2D_F}(\sigma\sqrt{LT})^{-1}\right\}$, then after $T$ rounds*

$$\mathbb{E}_{t \sim (Unif[T])}[\|\nabla F(w^t)\|_2^2] \leq \frac{2D_F L}{T} + \frac{2\sqrt{2}\sigma\sqrt{LD_F}}{\sqrt{T}} + DB,$$

*where* $$\sigma^2 = \max_{1 \leq t \leq T} 2\mathbb{E}[\|g^t(w^t) - \nabla F(w^t)\|_2^2] + 2 \max_{1 \leq t \leq T} \mathbb{E}_q[\|g^t(w^t) - \tilde{g}^t(w^t)\|_2^2], \tag{2}$$

*and $B = \max_{1 \leq t \leq T} \|\mathbb{E}_q[g^t(w^t) - \tilde{g}^t(w^t)]\|$. The expectation in the above equations is over the randomness in gradients and quantization.*

The above result relates the convergence of distributed SGD for non-convex functions to the *worst-case* mean square error (MSE) and bias in gradient mean estimates in Equation (2). Thus smaller the mean square error in gradient estimation, better convergence. Hence, we focus on the problem of distributed mean estimation (DME), where the goal is to estimate the mean of a set of vectors.

## 1.3 Differential privacy

While the above schemes reduce the communication cost, it is unclear what (if any) privacy guarantees they offer. We study privacy from the lens of differential privacy (DP). The notion of differential privacy [13] provides a strong notion of individual privacy while permitting useful data analysis in machine learning tasks. We refer the reader to [14] for a survey. Informally, for the output to be differentially private, the estimated model should be indistinguishable whether a particular client's data was taken into consideration or not. We define this formally in Section 2.

In the context of client based distributed learning, we are interested in the privacy of the gradients aggregated from clients; differential privacy for the average gradients implies privacy for the resulting model since DP is preserved by post-processing. The standard approach is to let the server add the noise to the averaged gradients (e.g., see [14, 2] and references within). However, the above only works under a restrictive assumption that the clients can trust the server. Our goal is to also minimize the need for clients to trust the central aggregator, and hence we propose the following model:

*Clients add their share of the noise to their gradients $g_i^t$ before transmission. Aggregation of gradients at the server results in an estimate with noise equal to the sum of the noise added at each client.*

This approach improves over server-controlled noise addition in several scenarios:

**Clients do not trust the server**: Even in the scenario when the server is not trustworthy, the above scheme can be implemented via cryptographically secure aggregation schemes [7], which ensures that the only information about the individual users the server learns is what can be inferred from the sum. Hence, differential privacy of the aggregate now ensures that the parameter server does not learn any individual user information. This will encourage clients to participate in the protocol even if they do not fully trust the server. We note that while secure aggregation schemes add to the communication cost (e.g., [7] adds $\log_2(k \cdot n)$ for $k$ levels of quantization), our proposed communication benefits still hold. For example, if $n = 1024$, a 4-bit quantization protocol would reduce communication cost by 67% compared to the 32 bit representation.

**Server is negligent, but not malicious**: the server may "forget" to add noise, but is not malicious and not interested in learning characteristics of individual users. However, if the server releases the learned model to public, it needs to be differentially-private.

A natural way to extend the results of [14, 2] is to let individual users add Gaussian noise to their gradients before transmission. Since the sum of Gaussians is Gaussian itself, differential privacy results follow. However, the transmitted values now are real numbers and the benefits of gradient compression are lost. Further, secure aggregation protocols [7] require discrete inputs. To resolve these issues, we propose that the clients add noise drawn from an appropriately parameterized Binomial distribution. We refer to this as the *Binomial mechanism*. Since Binomial random variables are discrete, they can be transmitted efficiently. Furthermore, the choice of the Binomial is convenient in the distributed setting because sum of Binomials is also binomially distributed i.e., if

$$Z_1 \sim \mathrm{Bin}(N_1, p), Z_2 \sim \mathrm{Bin}(N_2, p) \quad \text{then} \quad Z_1 + Z_2 \sim \mathrm{Bin}(N_1 + N_2, p).$$

Hence the total noise post aggregation can be analyzed easily, which is convenient for the distributed setting[2]. Binomial mechanism can be of independent interest in other applications with discrete output as well. Furthermore, unlike Gaussian it avoids floating point representation issues.

## 1.4 Summary of our results

**Binomial mechanism**: We first study Binomial mechanism as a generic mechanism to release discrete valued data. Previous analysis of the Binomial mechanism (where you add noise $\mathrm{Bin}(N, p)$) was due to [12], who analyzed the 1-dimensional case for $p = 1/2$ and showed that to achieve $(\varepsilon, \delta)$ differential privacy, $N$ needs to be $\geq 64 \log(2/\delta)/\varepsilon^2$. We improve the analysis in the following ways:

- $d$-**dimensions.** We extend the analysis of 1-dimensional Binomial mechanism to $d$ dimensions. Unlike the Gaussian distribution, Binomial is not rotation invariant making the analysis more involved. The key fact utilized in this analysis is that Binomial distribution is locally rotation-invariant around the mean.

- **Improvement.** We improve the previous result and show that $N \geq 8 \log(2/\delta)/\varepsilon^2$ suffices for small $\varepsilon$, implying that the Binomial and Gaussian mechanism perform identically as $\varepsilon \to 0$. We note that while this is a constant improvement , it is crucial in making differential privacy practical.

**Differentially-private distributed mean estimation (DME)**: A direct application of Gaussian mechanism requires $n \cdot d$ reals and hence $n \cdot d \cdot \log(nd)$ bits of communication. This can be prohibitive in practice. We first propose a direct application of quantization [33] and Binomial mechanism and characterize its privacy/error guarantees along with its communication costs. We further show that coupling the scheme with random rotation can significantly improve communication further. In particular, for $\varepsilon = O(1)$, we provide an algorithm achieving the same privacy and error tradeoff as that of the Gaussian mechanism with communication

$$ \leq n \cdot d \cdot \left( \log_2 \left( 1 + \frac{d}{n} \right) + \mathcal{O} \left( \log \log \left( \frac{nd}{\delta} \right) \right) \right) \text{ bits,} $$

per round of distributed SGD. Hence when $d \approx n$, the number of bits is $n \cdot d \cdot \log(\log(nd)/\delta)$.

The rest of the paper is organized as follows. In Section 2, we review the notion of differential privacy and state our results for the Binomial mechanism. Motivated by the fact that the convergence of SGD can be reduced to the error in gradient estimate computation per-round, we formally describe the problem of DME in Section 3 and state our results in Section 4.

In Section 4.2, we provide and analyze the implementation of the binomial mechanism in conjunction with quantization in the context of DME. The main idea is for each client to add noise drawn from an appropriately parameterized Binomial distribution to each quantized value before sending to the server. The server further subtracts the bias introduced by the noise to achieve an unbiased mean estimator. We further show in Section 4.3 that the rotation procedure proposed in [33] which reduces the MSE is helpful in reducing the additional error due to differential privacy.

## 2 Differential privacy

### 2.1 Notation

We start by defining the notion of differential privacy. Formally, given a set of data sets $\mathcal{D}$ provided with a notion of neighboring data sets $\mathcal{N}_{\mathcal{D}} \subset \mathcal{D} \times \mathcal{D}$ and a query function $f : \mathcal{D} \to \mathcal{X}$, a mechanism $\mathcal{M} : \mathcal{X} \to \mathcal{O}$ to release the answer of the query, is defined to be $(\varepsilon, \delta)$ differentially private if for any measurable subset $S \subseteq \mathcal{O}$ and two neighboring data sets $(D_1, D_2) \in \mathcal{N}_{\mathcal{D}}$,

$$ \Pr\left( \mathcal{M}(f(D_1)) \in S \right) \leq e^{\varepsilon} \Pr\left( \mathcal{M}(f(D_2)) \in S \right) + \delta. \tag{3} $$

Unless otherwise stated, for the rest of the paper, we will assume the output spaces $\mathcal{X}, \mathcal{O} \subseteq \mathbb{R}^d$. We consider the mean square error as a metric to measure the error of the mechanism $\mathcal{M}$. Formally,

$$ \mathcal{E}(\mathcal{M}) \triangleq \max_{D \in \mathcal{D}} \mathbb{E}[\|\mathcal{M}(f(D)) - f(D)\|_2^2]. $$

A key quantity in characterizing differential privacy for many mechanisms is the sensitivity of a query $f : \mathcal{D} \to \mathbb{R}^d$ in a given norm $\ell_q$. Formally this is defined as

$$ \Delta_q \triangleq \max_{(D_1, D_2) \in \mathcal{N}_{\mathcal{D}}} \|f(D_1) - f(D_2)\|_q. \tag{4} $$

The canonical mechanism to achieve $(\varepsilon, \delta)$ differential privacy is the Gaussian mechanism $\mathcal{M}_g^{\sigma}$ [14]: $\mathcal{M}_g^{\sigma}(f(D)) \triangleq f(D) + Z$, where $Z \sim \mathcal{N}(0, \sigma^2 \mathbb{I}_d)$. We now state the well-known privacy guarantee of the Gaussian mechanism.

**Lemma 1** ( [14]). *For any $\delta$, $\ell_2$ sensitivity bound $\Delta_2$, and $\sigma$ such that $\sigma \geq \Delta_2 \sqrt{2 \log 1.25/\delta}$, $\mathcal{M}_g^{\sigma}$ is $(\frac{\Delta_2}{\sigma}\sqrt{2 \log 1.25/\delta}, \delta)$ differentially private [3] and the error is bounded by $d \cdot \sigma^2$.*

## 2.2 Binomial Mechanism

We now define the Binomial mechanism for the case when the output space $\mathcal{X}$ of the query $f$ is $\mathbb{Z}^d$. The Binomial mechanism is parameterized by three quantities $N, p, s$ where $N \in \mathbb{N}, p \in (0, 1)$, and quantization scale $s = 1/j$ for some $j \in \mathbb{N}$ and is given by

$$\mathcal{M}_b^{N,p,s}(f(D)) \triangleq f(D) + (Z - Np) \cdot s, \tag{5}$$

where for each coordinate $i$, $Z_i \sim \text{Bin}(N, p)$ and independent. One dimensional binomial mechanism was introduced by [12] for the case when $p = 1/2$. We analyze the mechanism for the general $d$-dimensional case and for any $p$. This analysis is involved as the Binomial mechanism is not rotation invariant. By carefully exploiting the local rotation invariant structure near the mean, we show that:

**Theorem 1.** *For any $\delta$, parameters $N, p, s$ and sensitivity bounds $\Delta_1, \Delta_2, \Delta_\infty$ such that*

$$Np(1 - p) \geq \max\left(23 \log(10d/\delta), 2\Delta_\infty/s\right),$$

*the Binomial mechanism is $(\varepsilon, \delta)$ differentially private for*

$$\varepsilon = \frac{\Delta_2\sqrt{2\log\frac{1.25}{\delta}}}{s\sqrt{Np(1-p)}} + \frac{\Delta_2 c_p\sqrt{\log\frac{10}{\delta}} + \Delta_1 b_p}{sNp(1-p)(1-\delta/10)} + \frac{\Delta_\infty d_p \log\frac{1.25}{\delta} + \Delta_\infty d_p \log\frac{20d}{\delta}\log\frac{10}{\delta}}{sNp(1-p)}, \tag{6}$$

*where $b_p$, $c_p$, and $d_p$ are defined in (16), (11), and (15) respectively, and for $p = 1/2$, $b_p = 1/3$, $c_p = 5/2$, and $d_p = 2/3$. The error of the mechanism is $d \cdot s^2 \cdot Np(1 - p)$.*

The proof is given in Appendix B. We make some remarks regarding the design and the guarantee for the Binomial Mechanism. Note that the privacy guarantee for the Binomial mechanism depends on all three sensitivity parameters $\Delta_2, \Delta_\infty, \Delta_1$ as opposed to the Gaussian mechanism which only depends on $\Delta_2$. The $\Delta_1$ and $\Delta_\infty$ terms can be seen as the added complexity due to discretization.

Secondly setting $s = 1$ (i.e. providing no scale to the noise) in the expression (6), it can be readily seen that the terms involving $\Delta_1$ and $\Delta_2$ scale differently with respect to the variance of the noise. This motivates the use of the accompanying quantization scale $s$ in the mechanism. Indeed it is possible that the resolution of the integer that is provided by the Binomial noise could potentially be too large for the problem leading to worse guarantees. In this setting, the quantization parameter $s$ helps normalize the noise correctly. Further, it can be seen as long as the variance of the random variable $s \cdot Z$ is fixed, increasing $Np(1 - p)$ and decreasing $s$ makes the Binomial mechanism closer to the Gaussian mechanism. Formally, if we let $\sigma = s\sqrt{Np(1-p)}$ and $s \leq \sigma/(c\sqrt{d})$, then using the Cauchy-Schwartz inequality, the $\varepsilon$ guarantee (6) can be rewritten as

$$\varepsilon = (\Delta_2/\sigma)\sqrt{2\log 1.25/\delta}\left(1 + \mathcal{O}\left(1/c\right)\right).$$

The variance of the Binomial distribution is $Np(1 - p)$ and the leading term in $\varepsilon$ matches exactly the $\varepsilon$ term in Gaussian mechanism. Furthermore, if $s$ is $o(1/\sqrt{d})$, then this mechanism approaches the Gaussian mechanism. This result agrees with the Berry-Esseen type Central limit theorems for the convergence of one dimensional Binomial distribution to the Gaussian distribution. In Figure 1, we plot the error vs $\varepsilon$ for Gaussian and Binomial mechanism. Observe that as scale is reduced, error vs privacy trade-off for Binomial mechanism approaches that of Gaussian mechanism.

Finally note that, while $p = 1/2$ will in general be the optimal choice as it maximizes the variance for a fixed communication budget, there might be corner cases wherein the required variance is so small that it cannot be achieved by an integer choice of $N$ and $p = 1/2$. Our results working with general $p$ also cover these corner cases.

## 3 Distributed mean estimation (DME)

We have related the SGD convergence rate to the MSE in approximating the gradient at each step in Corollary 1. Eq. (1) relates the communication cost of SGD to the communication cost of estimating gradient means. Advanced composition theorem (Thm. 3.5 [19]) or moments accounting [2] can be used to relate the privacy guarantee of SGD to that of gradient mean estimate at each instance $t$. We also note that in SGD, we often sample the clients, standard privacy amplification results via sampling [2], can be used to get tighter bounds in this case.

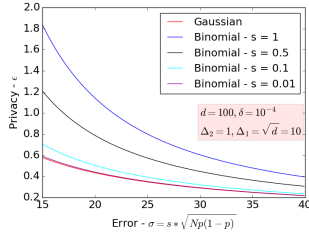

Figure 1: Comparison of error vs privacy for Gaussian and Binomial mechanism at different scales

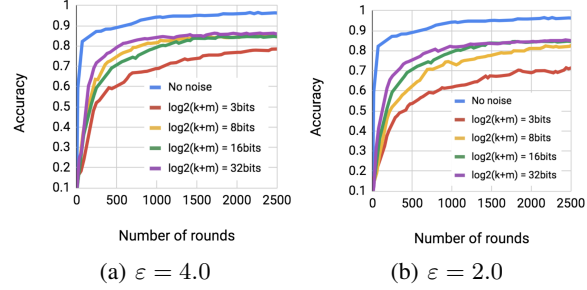

(a) $\varepsilon = 4.0$        (b) $\varepsilon = 2.0$

Figure 2: cpSGD with rotation on the infinite MNIST dataset. $k$ is the number of quantization levels, and $m$ is the parameter of the binomial noise ($p = 0.5$, $s = 1$). The baseline is without quantization and differential privacy. $\delta = 10^{-9}$.

Therefore, akin to [33], in the rest of the paper we just focus on the MSE and privacy guarantees of DME. The results for synchronous distributed GD follow from Corollary 1 (convergence), advanced composition theorem (privacy), and Eq. (1) (communication).

Formally, the problem of DME is defined as given $n$ vectors $X \triangleq \{X_1 \ldots X_n\}$ where $X_i \in \mathbb{R}^d$ is on client $i$, we wish to compute the mean $\bar{X} = \frac{1}{n} \sum_{i=1}^{n} X_i$ at a central server. For gradient descent at each round $t$, $X_i$ is set to $g_i^t$. DME is a fundamental building block for many distributed learning algorithms including distributed PCA/clustering [24].

While analyzing private DME we assume that each vector $X_i$ has bounded $\ell_2$ norm, i.e. $\|X_i\| \leq D$. The reason to make such an assumption is to be able to define and analyze the privacy guarantees and is often enforced in practice by employing gradient clipping at each client. We note that this assumption appears in previous works on gradient descent and differentially private gradient descent (e.g. [2]). Since our results also hold for all gradients without any statistical assumptions, we get desired convergence results and privacy results for SGD.

### 3.1 Communication protocol

Our proposed communication algorithms are simultaneous and independent, i.e., the clients independently send data to the server at the same time. We allow the use of both private and public randomness. Private randomness refers to random values generated by each client separately, and public randomness refers to a sequence of random values that are shared among all parties[4].

Given $n$ vectors $X \triangleq \{X_1 \ldots X_n\}$ where $X_i \in \mathbb{R}^d$ resides on a client $i$. In any independent communication protocol, each client transmits a function of $X_i$ (say $q(X_i)$), and a central server estimates the mean by some function of $q(X_1), q(X_2), \ldots, q(X_n)$. Let $\pi$ be any such protocol and let $\mathcal{C}_i(\pi, X_i)$ be the expected number of bits transmitted by the $i$-th client during protocol $\pi$, where throughout the paper, expectation is over the randomness in protocol $\pi$.

Let $\mathcal{C}_i(\pi, X_i)$ be the number of bits transmitted by client $i$. The total number of bits transmitted by all clients with the protocol $\pi$ is $\mathcal{C}(\pi, X_1^n) \stackrel{\text{def}}{=} \sum_{i=1}^{n} \mathcal{C}_i(\pi, X_i)$. Let the estimated mean be $\hat{\bar{X}}$. For a protocol $\pi$, the MSE of the estimate is $\mathcal{E}(\pi, X_1^n) = \mathbb{E}\left[\|\hat{\bar{X}} - \bar{X}\|_2^2\right]$. We note that bounds on $\mathcal{E}((\pi, X_1^n)$, translates to bounds on gradients estimates in Eq. (2) and result in convergence guarantees via Corollary 1.

### 3.2 Differential privacy

To state the privacy results for DME, we define the notion of data sets and neighbors as follows. A dataset is a collection of vectors $X = \{X_1, \ldots X_n\}$. The notion of neighboring data sets typically corresponds to those differing only on the information of one user, i.e. $X, X_{\otimes i}$ are neighbors if they differ in one vector. Note that this notion of neighbors for DME in the context of distributed gradient

descent translates to two data sets $F = f_1, f_2, \ldots f_n$ and $F' = f'_1, f'_2, \ldots f'_n$ being neighbors if they differ in one function $f_i$ and corresponds to guaranteeing privacy for individual client's data. The bound $\|X_i\|_2 \leq D$ translates to assuming $\|g_i^t\| \leq D$, ensured via gradient clipping.

# 4  Results for distributed mean estimation (DME)

In this section we describe our algorithms, the associated MSE, and the privacy guarantees in the context of DME. First, we first establish a baseline by stating the results for implementing the Gaussian mechanism by adding Gaussian noise on each client vector.

## 4.1  Gaussian protocol

In the Gaussian mechanism, each client sends vector $Y_i = X_i + Z_i$, where $Z_i$s are i.i.d distributed as $\mathcal{N}(0, \sigma^2 \mathbb{I}_d)$. The server estimates the mean by $\hat{\bar{X}} = 1/n \cdot \sum_{i=1}^{n} Y_i$. We refer to this protocol as $\pi_g$. Since $\sum_{i=1}^{n} Z_i/n$ is distributed as $\mathcal{N}(0, \sigma^2 \mathbb{I}_d/n)$ the above mechanism is equivalent to applying the Gaussian mechanism on the output with variance $\sigma^2/n$. Since changing any of the $X_i$'s changes the norm of $\bar{X}$ by at most $2D/n$, the following theorem follows directly from Lemma 1.

**Theorem 2.** *Under the Gaussian mechanism, the mean estimate is unbiased and communication cost is $n \cdot d$ reals. Moreover, for any $\delta$ and $\sigma \geq \frac{2D}{\sqrt{n}} \cdot \sqrt{2 \log 1.25/\delta}$, it is $(\varepsilon, \delta)$ differentially private for*

$$\varepsilon = \frac{2D}{\sqrt{n}\sigma} \sqrt{2 \log \frac{1.25}{\delta}} \quad and \quad \mathcal{E}(\pi_g, X) = \frac{d\sigma^2}{n},$$

We remark that real numbers can potentially be quantized to $\mathcal{O}(\log dn/\varepsilon\delta)$ bits with insignificant effect to privacy[5]. However this is asymptotic and can be prohibitive in practice [20], where we have a small fixed communication budget and $d$ is of the order of millions. A natural way to reduce communication cost is via quantization, where each client quantizes $Y_i$s before transmitting. However how privacy guarantees degrade as the quantization of the Gaussian mechanism is hard to analyze particularly under aggregation. Instead we propose to use the Binomial mechanism which we describe next.

## 4.2  Stochastic $k$-level quantization + Binomial mechanism

We now define the mechanism $\pi_{sk}(\text{Bin}(m, p))$ based on $k$-bit stochastic quantization $\pi_{sk}$ proposed in [33] composed with the Binomial mechanism. It will be parameterized by 3 quantities $k, m, p$.

First, the server sends $X^{\max}$ to all the clients, with the hope that for all $i, j$, $-X^{\max} \leq X_i(j) \leq X^{\max}$. The clients then clip each coordinate of their vectors to the range $[-X^{\max}, X^{\max}]$. For every integer $r$ in the range $[0, k)$, let $B(r)$ represent a bin (one for each $r$), i.e.

$$B(r) \stackrel{\text{def}}{=} -X^{\max} + \frac{2rX^{\max}}{k-1}, \tag{7}$$

The algorithm quantizes each coordinate into one of the bins stochastically and adds scaled Binomial noise. Formally client $i$ computes the following quantities for every $j$

$$U_i(j) = \begin{cases} B(r+1) & \text{w.p. } \frac{X_i(j)-B(r)}{B(r+1)-B(r)} \\ B(r) & \text{otherwise.} \end{cases} \qquad Y_i(j) = U_i(j) + \frac{2X^{\max}}{k-1} \cdot T_i(j). \tag{8}$$

where $r$ is such that $X_i(j) \in [B(r), B(r+1)]$ and $T_i(j) \sim \text{Bin}(m, p)$. The client sends $Y_i$ to the server. The server now estimates $\bar{X}$ by

$$\hat{\bar{X}}_{\pi_{sk}(\text{Bin}(m,p))} = \frac{1}{n} \sum_{i=1}^{n} \left( Y_i - \frac{2X^{\max}mp}{k-1} \right). \tag{9}$$

If $\forall j$, $X_i(j) \in [-X^{\max}, X^{\max}]$, then $\mathbb{E}\left[Y_i - \frac{2X^{\max}mp}{k-1}\right] = X_i$, and $\hat{\bar{X}}_{\pi_{sk}(\text{Bin}(m,p))}$ will be an unbiased estimate of the mean.

Before stating the formal guarantees we will require the definitions of the following quantities representing the sensitivity of the quantization protocol in the appropriate norm.

$$\Delta_\infty(X^{\max}, D) \overset{\text{def}}{=} k + 1$$

$$\Delta_1(X^{\max}, D) \overset{\text{def}}{=} \frac{\sqrt{d}D(k-1)}{X^{\max}} + \sqrt{\frac{2\sqrt{d}D\log(2/\delta)(k-1)}{X^{\max}}} + \frac{4}{3}\log\frac{2}{\delta}$$

$$\Delta_2(X^{\max}, D) \overset{\text{def}}{=} \frac{D(k-1)}{X^{\max}} + \sqrt{\Delta_1 + \sqrt{\frac{2\sqrt{d}D\log(2/\delta)(k-1)}{X^{\max}}}}. \tag{10}$$

For brevity of notation we have suppressed the parameters $k, \delta$ from the LHS. With no prior information on $X^{\max}$, the natural choice is to set $X^{\max} = D$. With this value of $X^{\max}$ we characterize the MSE, sensitivity, and communication complexity of $\pi_{sk}(\text{Bin}(m, p))$ below leveraging Theorem 1.

**Theorem 3.** *If $X^{\max} = D$, then the mean estimate is unbiased and*

$$\mathcal{E}(\pi_{sk}(\textbf{Bin}(m, p)), X^n) \leq \frac{dD^2}{n(k-1)^2} + \frac{d}{n} \cdot \frac{4mp(1-p)D^2}{(k-1)^2},$$

*Furthermore if*

$$mnp(1-p) \geq \max\left(23\log(10d/\delta), 2\Delta_\infty(D, X^{max})\right),$$

*then for any $\delta$, $\hat{\bar{X}}_{\pi_{sk}(Bin(m,p))}$ is $(\varepsilon, 2\delta)$ differentially private where $\varepsilon$ (as given by Theorem 1) is*

$$\varepsilon = \frac{\Delta_2\sqrt{2\log\frac{1.25}{\delta}}}{\sqrt{mnp(1-p)}} + \frac{\Delta_2 c_p\sqrt{\log\frac{10}{\delta}} + \Delta_1 b_p}{mnp(1-p)(1-\delta/10)} + \frac{\Delta_\infty d_p\log\frac{1.25}{\delta} + \Delta_\infty d_p\log\frac{20d}{\delta}\log\frac{10}{\delta}}{mnp(1-p)},$$

*with sensitivity parameters $\{\Delta_1(X^{max}, D), \Delta_2(X^{max}, D), \Delta_\infty(X^{max}, D)\}$ as defined in (10).*

*Furthermore,*

$$\mathcal{C}(\pi_{sk}(\textbf{Bin}(m, p)), X^n) = n \cdot (d\log_2(k + m) + \tilde{\mathcal{O}}(1)).[6]$$

We provide the proof in Appendix D. The first term in the expression for $\varepsilon$ in the above theorem recovers the same guarantee as that of the Gaussian mechanism (Theorem 2). Further, it can be seen that the trailing terms are negligible when $k >> \sqrt{d}$. Formally this leads to the following corollary summarizing the communication cost for $\varepsilon \leq 1$ for achieving the same guarantee as the Gaussian mechanism.

**Corollary 2.** *There exists an implementation of $\pi_{sk}(\textbf{Bin}(m, p))$, which achieves the same privacy and error as the full precision Gaussian mechanism with a total communication complexity of*

$$n \cdot d \cdot \left(\log_2\left(\sqrt{d} + \frac{d}{n\varepsilon^2}\right) + \mathcal{O}\left(\log\log\left(\frac{nd}{\varepsilon\delta}\right)\right)\right) \text{ bits.}$$

The communication cost of the above algorithm is $\Omega(\log d)$ bits per coordinate per client, which can be prohibitive. In the next section we show that these bounds can be further improved via rotation.

## 4.3 Error reduction via randomized rotation

As seen in Corollary 2, for $\pi_{sk}(\text{Bin}(m, p))$ to have error and privacy same as that of the Gaussian mechanism, the best bound on the communication cost guaranteed is $\Omega(\log(d))$ bits per coordinate irrespective of how large $n$ is. The proof reveals that this is due to the error being proportional to $O(d(X^{\max})^2/n)$. Therefore MSE reduces when $X^{\max}$ is small, e.g., when $X_i$ is uniform on the unit sphere, $X^{\max}$ is $\mathcal{O}\left(\sqrt{(\log d)/d}\right)$ (whp) [10]. [33] showed that the same effect can be observed by randomly rotating the vectors before quantization. Here we show that random rotation reduces the leading term in the error as well as improves the privacy guarantee.

Using public randomness, all clients and the central server generate a random orthogonal matrix $R \in \mathbb{R}^{d\times d}$ according to some known distribution. Given a protocol $\pi$ for DME which takes inputs

$X_1 \ldots X_n$, we define $\text{Rot}(\pi, R)$ as the protocol where each client $i$ first computes, $X_i' = R X_i$, and runs the protocol on $X_1', X_2', \ldots X_n'$. The server then obtains the mean estimate $\hat{\bar{X}}'$ in the rotated space using the protocol $\pi$ and then multiplies by $R^{-1}$ to obtain the coordinates in the original basis, i.e., $\hat{\bar{X}} = R^{-1} \hat{\bar{X}}'$.

Due to the fact that $d$ can be huge in practice, we need orthogonal matrices that permit fast matrix-vector products. Naive matrices that support fast multiplication such as block-diagonal matrices often result in high values of $\|X_i'\|_\infty^2$. Similar to [33], we propose to use a special type of orthogonal matrix $R = \frac{1}{\sqrt{d}} H A$, where $A$ is a random diagonal matrix with i.i.d. Rademacher entries ($\pm 1$ with probability 0.5) and $H$ is a Walsh-Hadamard matrix [18]. Applying both rotation and its inverse takes $\mathcal{O}(d \log d)$ time and $\mathcal{O}(1)$ space (with an in-place algorithm).

The next theorem provides the MSE and privacy guarantees for $\text{Rot}(\pi_{sk}(\text{Bin}(m,p)), HA)$.

**Theorem 4** (Appendix E). *For any $\delta$, let $X^{\max} = 2D \sqrt{\frac{\log(2nd/\delta)}{d}}$, then*

$$\mathcal{E}(Rot(\pi_{sk}(Bin(m,p))), HA) \leq \frac{2 \log \frac{2nd}{\delta} \cdot D^2}{n(k-1)^2} + \frac{8 \log \frac{2nd}{\delta} \cdot mp(1-p)D^2}{n(k-1)^2} + 4D^2\delta^2.$$

*The bias of mean estimate is bounded by $\leq 2D\delta$. Furthermore if*

$$mnp(1-p) \geq \max\left(23\log(10d/\delta), 2\Delta_\infty(D, X^{max})\right),$$

*then $\hat{\bar{X}}(Rot(\pi_{sk}(Bin(m,p))))$ is $(\varepsilon, 3\delta)$ differentially private where $\varepsilon$ (as given by Theorem 1) is*

$$\varepsilon = \frac{\Delta_2 \sqrt{2 \log \frac{1.25}{\delta}}}{\sqrt{mnp(1-p)}} + \frac{\Delta_2 c_p \sqrt{\log \frac{10}{\delta}} + \Delta_1 b_p}{mnp(1-p)(1-\delta/10)} + \frac{\Delta_\infty d_p \log \frac{1.25}{\delta} + \Delta_\infty d_p \log \frac{20d}{\delta} \log \frac{10}{\delta}}{mnp(1-p)},$$

*with sensitivity parameters $\{\Delta_1(X^{max}, D), \Delta_2(X^{max}, D), \Delta_\infty(X^{max}, D)\}$ (Eq. (10)). Furthermore,*

$$\mathcal{C}(Rot(\pi_{sk}(Bin(m,p))), X^n) = n \cdot (d \log_2(k+m) + \tilde{\mathcal{O}}(1)).$$

The following corollary now bounds the communication cost for $\text{Rot}(\pi_{sk}(\text{Bin}(m,p)), HA)$ when $\varepsilon \leq 1$ akin to Corollary 2.

**Corollary 3.** *There exists an implementation of $Rot(\pi_{sk}(Bin(m,p)), HA)$, that achieves the same error and privacy of the full precision Gaussian mechanism with a total communication complexity:*

$$n \cdot d \left( \log_2 \left(1 + \frac{d}{n\varepsilon^2}\right) + \mathcal{O}\left(\log\log \frac{dn}{\varepsilon\delta}\right)\right) \text{ bits.}$$

Note that $k$ is no longer required to be set to $\Omega(\sqrt{d})$ and hence if $d = o(n\varepsilon^2)$, then $\text{Rot}(\pi_{sk}(\text{Bin}(m,p)), HA)$ has the same privacy and utilities as the Gaussian mechanism, but with just $\mathcal{O}(nd \log\log(nd/\delta\varepsilon))$ communication cost.

## 5   Discussion

We trained a three-layer model (60 hidden nodes each with ReLU activation) on the infinite MNIST dataset [8] with 25M data points and 25M clients. At each step 10,000 clients send their data to the server. This setting is close to real-world settings of federated learning where there are hundreds of millions of users. The results are in Figure 2. Note that the models achieve different levels of accuracy depending on communication cost and privacy parameter $\varepsilon$. We note that we trained the model with exactly one epoch, so each sample was used at most once in training. In this setting, the per batch $\varepsilon$ and the overall $\varepsilon$ are the same.

There are several interesting future directions. On the theoretical side, it is not clear if our analysis of Binomial mechanism is tight. Furthermore, it is interesting to have better privacy accounting for Binomial mechanism via a moments accountant. On the practical side, we plan to explore the effects of neural network topology, over-parametrization, and optimization algorithms on the accuracy of the privately learned models.

## Footnotes

[1] $\eta$ is the per-coordinate quantization accuracy. To represent a $d$ dimensional vector $X$ to an constant accuracy in Euclidean distance, each coordinate is usually quantized to an accuracy of $\eta = 1/\sqrt{d}$.

[2]Another choice is the Poisson distribution. Different from Poisson, the Binomial distribution has bounded support and has an easily analyzable communication complexity which is always bounded.

[3] All logs are to base $e$ unless otherwise stated.

[4]Public randomness can be emulated by the server communicating a random seed

[5]Follows by observing that quantizing all values to $1/\text{poly}(n, d, 1/\varepsilon, \log 1/\delta)$ accuracy ensures minimum loss in privacy. In practice this is often implemented using 32 bits of quantization via float representation.

[6]$\tilde{\mathcal{O}}$ is used to denote poly-logarithmic factors.

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
