[Supplementary Material · cpSGD_Camera_Ready_NIPS_Supplementary.pdf]

## A  Proof of biased SGD

The proof is similar to the SGD proof of [16], however we account for bias in gradient estimates. Define the random variable $\delta_t \triangleq \tilde{g}_t(w_t) - \nabla F(w_{t-1})$. By the definitions of $L$ and $\gamma$,

$$
\begin{aligned}
F(w_{t+1}) - F(w_t) &\leq \nabla F(w_t)^T(w_{t+1} - w_t) + \frac{L}{2}\|w_{t+1} - w_t\|^2 \\
&\leq -\nabla F(w_t)^T(\gamma \tilde{g}_t(w_t)) + \gamma^2 \frac{L}{2}\|\tilde{g}_t(w_t)\|^2 \\
&\leq -\gamma(1 - \frac{\gamma L}{2})\|\nabla F(w_t)\|^2 + \gamma(1 - \gamma L)\|\nabla F(w_t)\|\|\delta_t\| + \gamma^2 \frac{L}{2}\|\delta_t\|^2,
\end{aligned}
$$

where the last inequality uses the fact that $\gamma L \leq 1$. Rearranging the above inequality and summing over all $t$ we get that

$$
\mathbb{E}_{t \in \text{Uniform}(T)}[\|\nabla F(w_t)\|^2]
$$

$$
\begin{aligned}
&\leq \frac{1}{T\gamma(2 - \gamma L)}\left(2(F(w_0) - F(w^*)) + T\gamma^2 L\mathbb{E}\|\delta_t\|^2\right) + \frac{2\gamma(1 - \gamma L)}{\gamma(2 - \gamma L)}\left(\frac{1}{T}\sum_{t=1}^{T}\|\nabla F(w_t)\|\|\mathbb{E}[\delta_t]\|\right) \\
&\leq \frac{1}{T\gamma(2 - \gamma L)}\left(2D_F + T\gamma^2 L\sigma^2\right) + \frac{2\gamma(1 - \gamma L)}{\gamma(2 - \gamma L)}DB \\
&\leq \frac{2D_F}{T}\max\left\{L, \frac{\sigma\sqrt{LT}}{\sqrt{2D_F}}\right\} + \frac{\sigma\sqrt{2LD_F}}{\sqrt{T}} + DB \\
&\leq \frac{2D_F L}{T} + \frac{2\sqrt{2LD_F}\sigma}{\sqrt{T}} + DB.
\end{aligned}
$$

## B  Binomial Mechanism - Proof of Theorem 1

To remind the reader, the binomial mechanism for releasing discrete valued queries on a database is defined as follows. Given a set of databases $\mathcal{D}$ and an integer valued query $f : \mathcal{D} \to \mathbb{Z}^d$, the binomial mechanism samples a vector $Z \in \mathbb{Z}^d$ such that all its coordinates are distributed as the binomial distribution with parameters $N, p$, i.e.

$$
Z(j) \sim \text{Bin}(N, p)
$$

The Binomial mechanism releases the vector $s(Z - Np) + f(D)$ as the output to the query. For the analysis the reader is referred to the definition of $\ell_q$ norm sensitivity $\Delta_q$ for any $q > 0$ defined in (4). The $q$ of interest to us for the Binomial mechanism will be $q = \{1, 2, \infty\}$. Since our requirement from the Binomial mechanism will be symmetric w.r.t. $p$ and $1 - p$, throughout this proof, we assume that $p \leq 1/2$.

To prove Theorem 1, we need few auxiliary lemmas. We first state two inequalities which we use through-out the proof.

**Lemma 2** (Bernstein's inequality)**.** *Let $X_1, X_2 \ldots X_n$ be independent random variables such that $E[X_i] = 0$ and $|X_i| \leq M$ w.p. 1. Let $\sigma_i^2 \triangleq \mathbb{E}[X_i^2]$. Then for any $\delta \geq 0$,*

$$
Pr\left(\sum X_i \geq \sqrt{2\sum \sigma_i^2 \log\frac{1}{\delta}} + \frac{2}{3} \cdot M \log\frac{1}{\delta}\right) \leq \delta.
$$

**Lemma 3** (Efron-Stein inequality)**.** *Let $f$ be a symmetric function of $n$ independent random variables $X_1, X_2, \ldots X_n$. Let $X_1'$ be an i.i.d. copy of $X_1$, then*

$$
Var(f) \leq \frac{n}{2} \cdot \mathbb{E}\left[(f(X_1, X_2, \ldots X_n) - f(X_1', X_2, \ldots X_n))^2\right].
$$

We use the above two results in the next two lemmas.

**Lemma 4.** *Let $T \sim Bin(N, p)$, $i \in [0, N]$, $t \in \mathbb{Z}$, $i - t \in [0, N]$. Then*

$$
\frac{\Pr(T = i - t)}{\Pr(T = i)} \leq \exp\left(t \cdot \log\frac{(i+1)(1-p)}{(N - i + 1)p}\right)
$$

*Proof.*

$$\frac{\Pr(T = i - t)}{\Pr(T = i)} \triangleq \frac{\binom{N}{i-t}}{\binom{N}{i}} \frac{p^{i-t}(1-p)^{N-i+t}}{p^i(1-p)^{N-i}}$$

$$= \frac{i!(N-i)!}{(i-t)!(N-i+t)!}\left(\frac{1-p}{p}\right)^t$$

$$\leq \left(\frac{(i+1)(1-p)}{(N-i+1)p}\right)^t,$$

where the inequality follows from considering the two cases when $t$ can be positive or negative. $\quad\square$

**Lemma 5.** *Let $t_1, t_2, \ldots t_d$ be $d$ real numbers. Let $v_i \sim Bin(N, p)$ independently such that $Np(1 - p) \geq 39$. Let $A$ be the event that $\|v_i - Np\|_\infty \leq \beta$ for some $\beta$, such that $\beta \leq N \min(p, 1 - p)/3$. Then for any $\delta$, with probability $\geq 1 - \delta$ conditioned on $A$,*

$$\sum_{i=1}^d t_i \left( \cdot \log \frac{(v_i+1)(1-p)}{(N-v_i+1)p} - \frac{v_i+1}{Np} + \frac{N-v_i+1}{N(1-p)} \right)$$

$$\leq \frac{2\|t\|_1(p^2+(1-p)^2)}{3Np(1-p)(\Pr(A))} + \frac{\|t\|_2 c_p}{Np(1-p)\sqrt{\Pr(A)}} \cdot \sqrt{\log \frac{1}{\delta}} + \frac{4\|t\|_\infty(\beta+1)^2(p^2+(1-p)^2)}{9N^2p^2(1-p)^2}\log\frac{1}{\delta},$$

*where $c_p$ is given by*

$$c_p \triangleq \sqrt{2}(3p^3 + 3(1-p)^3 + 2p^2 + 2(1-p)^2). \tag{11}$$

*Proof.* Since $\beta \leq N\min(p, 1-p)/3$ and for any $z \geq -1/3$, $|\log(1+z) - z| \leq 1.95z^2/3$,

$$\left|\log \frac{(v_i+1)(1-p)}{(N-v_i+1)p} - \frac{v_i+1}{Np} + \frac{N-v_i+1}{N(1-p)}\right|$$

$$\leq \frac{1.95}{3}\left|\frac{v_i+1-Np}{Np}\right|^2 + \frac{1.95}{3}\left|\frac{N-v_i+1-N-Np}{N(1-p)}\right|^2.$$

Hence we can bound the expectation as

$$\mathbb{E}\left[\log \frac{(v_i+1)(1-p)}{(N-v_i+1)p} - \frac{v_i+1}{Np} + \frac{N-v_i+1}{N(1-p)}\Big|A\right]$$

$$\leq \mathbb{E}\left[\frac{1.95}{3}\left|\frac{v_i+1-Np}{Np}\right|^2 + \frac{1.95}{3}\left|\frac{N-v_i+1-N-Np}{N(1-p)}\right|^2\Big|A\right]$$

$$\overset{(a)}{\leq} \frac{1}{\Pr(A)} \cdot \mathbb{E}\left[\frac{1.95}{3}\left|\frac{v_i+1-Np}{Np}\right|^2 + \frac{1.95}{3}\left|\frac{N-v_i+1-N-Np}{N(1-p)}\right|^2\right]$$

$$\overset{(b)}{\leq} \frac{1}{\Pr(A)} \cdot \frac{2(p^2+(1-p)^2)}{3Np(1-p)},$$

Where $(a)$ uses the fact that for any positive random variable $X$ and any event $A$, $\mathbb{E}[X] \geq \Pr(A)\mathbb{E}[X|A]$. $(b)$ uses the fact that $Np(1 - p) \geq 39$. Note that the function we are considering is a sum of functions of $d$ independent binomial random variables and hence we can apply Bernstein' inequality. To this end, we bound $\sigma_i^2$ and $M$. Since $\|v_i - Np\|_\infty$ is bounded,

$$\left|\log \frac{(v_i+1)(1-p)}{(N-v_i+1)p} - \frac{v_i+1}{Np} + \frac{N-v_i+1}{N(1-p)}\right| \leq \frac{2}{3}\left|\frac{v_i+1-Np}{Np}\right|^2 + \frac{2}{3}\left|\frac{N-v_i+1-N-Np}{N(1-p)}\right|^2$$

$$\leq \frac{2}{3}\frac{(\beta+1)^2(p^2+(1-p))^2}{N^2p^2(1-p)^2},$$

where the first inequality follows from the fact that $\beta \leq N\min(p, 1-p)/3$ and for any $z \geq -1/3$, $|\log(1+z) - z| \leq 2z^2/3$. Hence we can set $M = \frac{2}{3}\frac{(\beta+1)^2(p^2+(1-p))^2}{N^2p^2(1-p)^2}$. We now bound the variance:

$$\mathrm{Var}\left(\sum_{i=1}^d t_i \cdot \log \frac{(v_i+1)(1-p)}{(N-v_i+1)p} - \frac{v_i+1}{Np} + \frac{N-v_i+1}{N(1-p)}\Big|A\right).$$

We now bound $\sigma_i^2$. Observe that the term corresponding to $i$, is a function of $n$ independent Bernoulli $p$ random variables $X_i(j)$, for $1 \le j \le d$. We bound the expected square change in the function for any of these variables $X_i(j)$ and then use Efron-Stein inequality. Let $\mathbb{E}_A$ denote the expectation conditioned on the event $A$. Without loss of generality we first consider the contribution of the term $X_i(j)$. Let $w = \sum_{j' \ne j}^{n} X_i(j')$, then

$$\mathbb{E}_A \left[ t_i \cdot \log \frac{(v_i+1)(1-p)}{(N-v_i+1)p} - \frac{v_i+1}{Np} + \frac{N-v_i+1}{N(1-p)} - t_i \cdot \log \frac{(v_i'+1)(1-p)}{(N-v_i'+1)p} - \frac{v_i'+1}{Np} + \frac{N-v_i'+1}{N(1-p)} \right]^2$$

$$= t_i^2 \mathbb{E}_A \left[ \cdot \log \frac{(w+X_i(j)+1)(1-p)}{(N-w-X_i(j)+1)p} - \frac{w+X_i(j)+1}{Np} + \frac{N-w-X_i(j)+1}{N(1-p)} \right]$$

$$- t_i^2 \mathbb{E}_A \left[ \cdot \log \frac{(w+X_i'(j)+1)(1-p)}{(N-w-X_i'(j)+1)p} - \frac{w+X_i'(j)+1}{Np} + \frac{N-w-X_i'(j)+1}{N(1-p)} \right]^2$$

$$\overset{(a)}{=} 2t_i^2 p(1-p) \mathbb{E}_A \left[ \log \left( 1 + \frac{1}{w+1} \right) + \log \left( 1 + \frac{1}{N-w} \right) - \frac{1}{Np} - \frac{1}{N(1-p)} \right]^2$$

$$\overset{(b)}{\le} 2t_i^2 p(1-p) \mathbb{E} \left[ \log \left( 1 + \frac{1}{w+1} \right) + \log \left( 1 + \frac{1}{N-w} \right) - \frac{1}{Np} - \frac{1}{N(1-p)} \right]^2 \cdot \frac{1}{\Pr(A)}$$

$$= 2t_i^2 p(1-p) \mathbb{E} \left[ \log \left( 1 + \frac{1}{w+1} \right) + \log \left( 1 + \frac{1}{N-w} \right) - \frac{1}{Np(1-p)} \right]^2 \cdot \frac{1}{\Pr(A)},$$

where $(a)$ uses the fact that the term is non-zero only if $X_i(j) = 1, X_i'(j) = 0$ or $X_i(j) = 0, X_i'(j) = 1$ and the probability of this event is $2p(1-p)$. $(b)$ uses the fact that for any positive random variable $X$ and any event $A$, $\mathbb{E}[X] \ge \Pr(A)\mathbb{E}[X|A]$. We first upper bound the term inside the expectation:

$$\left( \log \left( 1 + \frac{1}{w+1} \right) + \log \left( 1 + \frac{1}{N-w} \right) - \frac{1}{Np(1-p)} \right)^2$$

$$= \left( \log \left( 1 + \frac{1}{w+1} \right) + \log \left( 1 + \frac{1}{N-w} \right) \right)^2 + \frac{1}{N^2 p^2 (1-p)^2} -$$

$$\frac{2}{Np(1-p)} \left( \log \left( 1 + \frac{1}{w+1} \right) + \log \left( 1 + \frac{1}{N-w} \right) \right)$$

$$\le \frac{1}{(w+1)^2} + \frac{1}{(N-w)^2} + \frac{2}{w(N-w)} + \frac{1}{N^2 p^2 (1-p)^2} -$$

$$\frac{2}{Np(1-p)} \left( \frac{1}{w+1} - \frac{1}{2(w+1)^2} + \frac{1}{N-w} - \frac{1}{2(N-w)^2} \right)$$

$$= \frac{1}{(w+1)(w+2)} - \frac{2}{Np(1-p)} \frac{1}{w+1} + \frac{1}{(N-w)(N-w+1)} - \frac{2}{Np(1-p)} \frac{1}{N-w}$$

$$+ \frac{2}{w(N-w)}$$

$$+ \frac{1}{(w+1)^2(w+2)} + \frac{1}{(N-w)^2(N-w+1)} + \frac{1}{Np(1-p)} \left( \frac{1}{(w+1)^2} + \frac{1}{(N-w)^2} \right)$$

$$+ \frac{1}{N^2 p^2 (1-p)^2},$$

where the inequality uses the fact that for any positive $x$, $x - x^2/2 \le \log x \le x$. Observe that $w \sim \text{Bin}(n-1, p)$ and $N-1-w \sim \text{Bin}(n-1, 1-p)$. We use the following three inequalities, to bound the expectation of the term above. Similar results apply for $N-w$ as $N-1-w \sim \text{Bin}(n-1, 1-p)$. Since $1/w$ and $1/(N-w)$ are negatively correlated,

$$\mathbb{E} \left[ \frac{1}{w(N-w)} \right] \le \mathbb{E} \left[ \frac{1}{w} \right] \cdot \mathbb{E} \left[ \frac{1}{N-w} \right].$$

Furthermore, for any $i$

$$\mathbb{E} \left[ \frac{w!}{(w+i)!} \right] \le \frac{1}{(Np)^i}$$

and if $Np(1-p) \geq 2$,

$$\mathbb{E}\left[\frac{1}{(w+1)(w+2)} - \frac{2}{Np(1-p)}\frac{1}{w+1}\right] \leq \frac{1}{(Np)^2} - \frac{2}{N^2p^2(1-p)}.$$

Combining the above results and simplifying the terms, we get that the expectation of the required quantity is bounded by

$$= \frac{1}{N^3p^3(1-p)^3} \cdot (3p^3 + 3(1-p)^3 + 2p^2 + 2(1-p)^2).$$

Hence $\sigma_i^2$ is bounded by

$$\frac{1}{\Pr(A)} \cdot \frac{t_i^2}{N^2p^2(1-p)^2} \cdot (3p^3 + 3(1-p)^3 + 2p^2 + 2(1-p)^2),$$

and the lemma follows by Bernstein's inequality. $\qquad\square$

*Proof of Theorem 1.* Firstly note that it is sufficient to consider the differential privacy of the quantity $\frac{f(D)}{s} + Z$ where $Z$ is a Binomial random variable. Note that since $s$ is defined to be $1/j$ for some integer $j$ the output $f(D)/s$ remains integral. Further note that in this setting the $l_q$ norm sensitivity scales $\Delta_q/s$. The above reduction shows that the scale $s$ can be considered to be 1 in the rest of the proof.

Consider any two neighboring data sets $D_1, D_2$ and let $\Delta \triangleq f(D_2) - f(D_1)$. Note that showing the $(\varepsilon, \delta)$ differential privacy of the Binomial mechanism is equivalent to showing the following. Let $T$ be a vector such that $T(j) \sim \mathrm{Bin}(N, p)$ then for any vector $v \in [N]^d$ we have that

$$\Pr(T = v) \leq e^\varepsilon \Pr(T = v - \Delta) + \delta$$

To show the above we will first define a set $V$ such that

$$\Pr(T \in V) \geq 1 - \delta,$$

and for every element $v \in V$,

$$\Pr(T = v) \leq e^\varepsilon \Pr(T = v - \Delta).$$

Define $V$ as follows: $v \in V$ if and only if,

$$\|v - Np\|_\infty \leq \beta \triangleq \sqrt{2Np(1-p)\log(20d/\delta)} + \frac{2}{3}\max(p, 1-p)\log\frac{20d}{\delta}. \tag{12}$$

$$|\Delta \cdot (v - Np)| \leq \|\Delta\|_2\sqrt{2Np(1-p)\log(1.25/\delta)} + \frac{2}{3}\log(1.25/\delta)\|\Delta\|_\infty. \tag{13}$$

$$\forall j, \ v(j) - \Delta(j) \in [0, N] \text{ and } v(j) \in Np \pm Np(1-p)/3.$$

$$\sum_{i=1}^{d} \Delta(j) \cdot \left(\log\frac{(v(j)+1)(1-p)}{p(N-v(j)+1)} - \frac{v(j)+1}{Np} + \frac{N - v(j) + 1}{N(1-p)}\right) \leq \frac{2\|\Delta\|_1(p^2 + (1-p)^2)}{3Np(1-p)(1 - \delta/10)}$$

$$+ \frac{\|\Delta\|_2 c_p}{Np(1-p)\sqrt{1 - \delta/10}} \cdot \sqrt{\log\frac{10}{\delta}} + \frac{4\|\Delta\|_\infty(\beta+1)^2(p^2 + (1-p)^2)}{9N^2p^2(1-p)^2}\log\frac{10}{\delta}. \tag{14}$$

We will first show that the probability of this event is large.

The first condition follows from Bernstein's inequality with probability $\geq 1 - \delta/10$. For the second condition, observe that $\Delta \cdot (s - Np)$ is a function of $Nd$ independent random variables. A direct application of Bernstein's inequality yields that Equation (13) holds with probability $\geq 1 - \delta/1.25$. The third condition follows from the first condition as $\|\Delta\|_\infty \leq Np - \beta$ and $Np(1-p)/3 \geq \beta$. Applying Lemma 5 with $A$ being event that $\|v - Np\|_\infty \leq \beta$ and $\delta = \delta/10$, yields that the fourth equation holds with probability at least $1 - \delta/10$. Hence, by the union bound,

$$\Pr(T \notin V) \leq \delta.$$

We now prove the ratio of probabilities. For any $v$,

$$\frac{\Pr(T = v - \Delta)}{\Pr(T = v)}$$

$$= \prod_{i=1}^{d} \frac{\Pr(T(j) = v(j) - \Delta(j))}{\Pr(T(j) = v(j))}$$

$$\leq \exp\left(\sum_{i=1}^{d} \Delta(j) \cdot \log \frac{(v(j)+1)(1-p)}{p(N-v(j)+1)}\right)$$

$$= \exp\left(\sum_{i=1}^{d} \frac{\Delta(j)(v(j)-Np)}{Np(1-p)} + \sum_{i=1}^{d} \Delta(j) \cdot \left(\log \frac{(v(j)+1)(1-p)}{p(N-v(j)+1)} - \frac{v(j)+1}{Np} + \frac{N-v(j)+1}{N(1-p)}\right)\right.$$
$$\left. + \frac{\sum_{j=1}^{d} \Delta(j)(1-2p)}{Np(1-p))}\right)$$

where the inequality follows from Lemma 4. Since $v \in V$, applying Equations (12), (13), (14), together with the fact that $\beta \leq \sqrt{2.5Np(1-p)\log(20d/\delta)}$ (by the assumptions in the theorem) yields the following bound on the exponent.

$$\|\Delta\|_2 \cdot \sqrt{\frac{2\log\frac{1.25}{\delta}}{Np(1-p)}} + \frac{2\|\Delta\|_\infty}{3Np(1-p)}\log\frac{1.25}{\delta} + \frac{\|\Delta\|_2 c_p \sqrt{\log\frac{10}{\delta}}}{Np(1-p)\sqrt{1-\delta/10}} + \frac{\|\Delta\|_\infty d_p \log\frac{20d}{\delta}\log\frac{10}{\delta}}{Np(1-p)}$$
$$+ \frac{b_p \|\Delta\|_1}{Np(1-p)(1-\delta/10)},$$

where $c_p$ is defined in Equation (11) and

$$d_p \triangleq \frac{4}{3} \cdot (p^2 + (1-p)^2) \tag{15}$$

and

$$b_p \triangleq \frac{2(p^2 + (1-p)^2)}{3} + (1-2p). \tag{16}$$

$\square$

## C  High probability sensitivity

To describe our main lemma formally we need the following definition. Let $Q \triangleq \{q_i \in \mathbb{N}\}$ represent a set of natural numbers and $\Delta_Q \triangleq \{\Delta_{q_i}\}$ represent a subset of real numbers. For two random vectors $v_1, v_2$, the event $\|v_1 - v_2\|_Q \leq \Delta_Q$ is defined as

$$(\|v_1 - v_2\|_Q \leq \Delta_Q) \triangleq \bigcup_i (\|v_1 - v_2\|_{q_i} \leq \Delta_{q_i})$$

**Definition 1** (($\Delta_Q, \delta$) sensitivity). *Given a set of integers $Q$ and values $\Delta_Q, \delta$, we call a randomized function $f : \mathcal{D} \to \mathcal{X}$, ($\Delta_Q, \delta$) sensitive, if for any two neighboring data sets $D_1, D_2 \in \mathcal{N}_\mathcal{D}$, there exist coupled random variables $X_1, X_2 \in \mathcal{X}$ such that the marginal distributions of $X_1, X_2$ are identical to that of $f(D_1)$ and $f(D_2)$ and*

$$\Pr_{X_1, X_2} (\|X_1 - X_2\|_Q \leq \Delta_Q) \geq 1 - \delta. \tag{17}$$

We show the following result for high-probability sensitivity and the proof is provided in Appendix C.

**Lemma 6.** *Let $\mathcal{M} : \mathcal{X} \to \mathcal{O}$ be an ($\varepsilon, \delta$) differentially private mechanism for sensitivity $\Delta_Q$ and let $f : \mathcal{D} \to \mathcal{X}$ be a ($\Delta_Q, \delta'$) sensitive function. Then the composed mechanism $\mathcal{M}(f(D))$ is ($\varepsilon, \delta + \delta'$) differentially private.*

*Proof.* To show $(\varepsilon, \delta + \delta')$ differential privacy we need to show that for any two neighboring data sets $D_1, D_2$ and $O \subseteq \mathcal{O}$,

$$\Pr(\mathcal{M}(f(D_1)) \in O) \le e^{\varepsilon} \Pr(\mathcal{M}(f(D_2)) \in O) + \delta + \delta'.$$

Given any two neighboring data sets $D_1, D_2$ let $\Pr_{\Delta_Q,\delta}(X_1, X_2)$ represent the joint distribution of the coupled random variables $X_1, X_2$ guaranteed by Definition 1. Now for any $O \in \mathcal{O}$ we have that

$$
\begin{aligned}
Pr(\mathcal{M}(f(D_1)) \in O) &\triangleq \int_{s \in \mathcal{S}} \Pr(f(D_1) = s) \Pr(\mathcal{M}(s) \in O) \\
&\stackrel{(a)}{=} \left( \int_{s_1, s_2 \| \|s_1 - s_2\|_Q \le \Delta_Q} \Pr_{\Delta_Q,\delta}(s_1, s_2)(\Pr(\mathcal{M}(s_2) \in O)) \right) \\
&\qquad + \left( \int_{s_1, s_2 \| \|s_1 - s_2\|_Q \ge \Delta_Q} \Pr_{\Delta_Q,\delta}(s_1, s_2)(\Pr(\mathcal{M}(s_2) \in O)) \right) \\
&\stackrel{(b)}{=} \left( \int_{s_1, s_2 \| \|s_1 - s_2\|_Q \le \Delta_Q} \Pr_{\Delta_Q,\delta}(s_1, s_2)(\Pr(\mathcal{M}(s_2) \in O)) \right) + \delta \\
&\stackrel{(c)}{\le} \left( \int_{s_1, s_2 \| \|s_1 - s_2\|_Q \le \Delta_Q} \Pr_{\Delta_Q,\delta}(s_1, s_2)(e^{\varepsilon} \Pr(\mathcal{M}(s_2) \in O) + \delta) \right) + \delta' \\
&\stackrel{(d)}{\le} e^{\varepsilon} \left( \int_{s \in \mathcal{S}} \Pr(f(D_2) = s) \Pr(\mathcal{M}(s) \in O) \right) + \delta + \delta' \\
&\triangleq e^{\varepsilon} \Pr(\mathcal{M}(f(D_2)) \in O) + \delta + \delta'.
\end{aligned}
$$

In the above $(a), (d)$ follow from the fact that $\Pr_{\Delta_q,\delta}$ is a coupling, $(b)$ follows from the condition (17) guaranteed by the coupling and $(c)$ follows from the $(\varepsilon, \delta)$ differential privacy guarantee of the mechanism $\mathcal{M}$. $\qquad \square$

## D Application of Binomial Mechanism to Distributed Mean Estimation - Proof of Theorem 3

*Proof of Theorem 3.* We refer the readers to the definition of the protocol (Section 4.2) and in particular the definitions of the random variables $U_i, T_i$, and the estimator $\hat{\bar{X}}_{\pi_{sk}(\mathrm{Bin}(m,p))}$ given in equations (8) and (9) respectively.

The communication complexity follows immediately by noting that the protocol only transmits integers in the range $[0, k + m)$ and therefore only needs $\log(k + m)$ bits. We now prove the bound on the Mean Square Error of the protocol and then prove the sensitivity guarantee.

**Mean Square Error**

$$
\begin{aligned}
\|\hat{\bar{X}} - \bar{X}\|_2^2 &= \frac{1}{n^2} \sum_{j=1}^{d} \sum_{i=1}^{n} \mathbb{E}[(\hat{\bar{X}}_i(j) - X_i(j))^2] \\
&\le \frac{1}{n^2} \sum_{j=1}^{d} \sum_{i=1}^{n} \mathbb{E}\left[ \left( \frac{2X^{\max}}{k-1} \right)^2 (\mathrm{Var}(\mathrm{Ber}(p_i(j))) + \mathrm{Var}(\mathrm{Bin}(mp))) \right] \\
&\le (2X^{\max})^2 \left( \frac{d}{4n(k-1)^2} + \frac{d}{n^2} \frac{mnp(1-p)}{(k-1)^2} \right),
\end{aligned}
$$

where the equality follows from the fact that $\hat{\bar{X}}_i(j)$ are independent of each other and $\hat{\bar{X}}$ is an unbiased estimator of $\hat{\bar{X}}$. Setting $m, p, k$ as defined in the theorem proves the bound on MSE.

**Differential Privacy**

Given two neighboring data sets $X \triangleq \{X_1 \ldots X_n\}$ and $X_{\otimes n} \triangleq \{X_1' \ldots X_n'\}$ (where $X_i' = X_i$

for $i \in [1, n-1]$) we will first provide a high probability bound on the $\ell_1, \ell_2, \ell_\infty$ sensitivity of quantization protocol $\pi_{sk}$. In particular the following lemma provides the high probability sensitivity bounds.

**Lemma 7.** *For every $\delta$, given two neighboring data sets $X \triangleq \{X_1 \ldots X_n\}$ and $X_{\otimes n} \triangleq \{X_1' \ldots X_n'\}$ (where $X_i' = X_i$ for $i \in [1, n-1]$) we have that the protocol $\pi_{sk}$ is $(\{\Delta_1, \Delta_2, \Delta_\infty\}, \delta)$-sensitive (c.f. Definition 1) where $\Delta_1, \Delta_2, \Delta_\infty$ satisfy the following equations.*

$$\Delta_\infty \leq \frac{\|X_n - X_n'\|_\infty}{2X^{\max}/(k-1)} + 2 \tag{18}$$

$$\Delta_1 \leq \frac{\|X_n - X_n'\|_1}{2X^{\max}/(k-1)} + \sqrt{2\frac{\|X_n - X_n'\|_1 \log(2/\delta)}{2X^{\max}/(k-1)}} + \frac{4}{3}\log(2/\delta) \tag{19}$$

$$\Delta_2 \leq \frac{\|X_n - X_n'\|_2}{2X^{\max}/(k-1)} + \sqrt{\frac{\|X_n - X_n'\|_1}{2X^{\max}/(k-1)} + \sqrt{\frac{8\|X_n - X_n'\|_1 \log(2/\delta)}{2X^{\max}/(k-1)}} + \frac{4}{3}\log(2/\delta)}. \tag{20}$$

Further we note that the protocol $\pi_{sk}(\text{Bin}(m, p))$ is a composition of the binomial mechanism and the protocol $\pi_{sk}$. A direct application of Theorem 1 and Lemma 6 gives us that the mechanism $\pi_{sk}(\text{Bin}(m, p))$ is $(\varepsilon, 2\delta)$ differentially private for any $\delta \in (0, 1)$ and $\varepsilon$ satisfying the below conditions. [7] Note that the conditions required by Theorem 1 can be verified from the given conditions in Theorem 3. $\qquad \square$

We now provide a proof of Lemma 7.

*Proof of Lemma 7.* To this end we recall the definition of the random variables $U_i(j)$. Given $X^{\max}$ and $X^{\min}$ we associate to every integer $r$ in $[0, k)$ a bin $B(r)$ defined as

$$B(r) \triangleq -X^{\max} + \frac{2rX^{\max}}{k-1}$$

Further given a number $X \in [-X^{\max}, X^{\max}]$, let $r(X)$ be the integer such that $X \in [B(r(X)), B(r(X)+1)]$. We can now define the random variable

$$U(X) = \begin{cases} r(X) + 1 & \text{w.p. } \frac{X - B(r(X))}{B(r(X)+1) - B(r(X))} \\ r(X) & \text{otherwise.} \end{cases}$$

Now define the random variables $U_i^X(j) \triangleq U(X_i(j))$ and similarly $U_i^{X_{\otimes n}}(j) \triangleq U(X_i'(j))$. To provide high probability sensitivity bounds in accordance with Lemma 6, we need to define a coupling between the random variables $\sum_i U_i^X$ and $\sum_i U_i^{X_{\otimes n}}$. To do the above we will define a coupling between the random variables $U_i^X(j)$ and $U_i^{X_{\otimes n}}(j)$. The coupled random variables will be sampled as follows.

The defined coupling will have two cases. Define the set $S = \{(i, j) | r(X_i(j)) = r(X_i'(j))\}$. We first consider the case when $(i, j) \in S$. In this case we sample a random variable $\alpha_{ij} \in [0, 1]$ uniformly at random and define the random variables

$$Y_i(j) = \begin{cases} r(X_i(j)) + 1 & \text{if } \alpha_{ij} \leq \frac{X_i(j) - B(r(X_i(j)))}{B(r(X_i(j))+1) - B(r(X_i(j)))} \\ r(X_i(j)) & \text{otherwise.} \end{cases}$$

$$Y_i^{\otimes n}(j) = \begin{cases} r(X_i'(j)) + 1 & \text{if } \alpha_{ij} \leq \frac{X_i'(j) - B(r(X_i'(j)))}{B(r(X_i'(j))+1) - B(r(X_i'(j)))} \\ r(X_i'(j)) & \text{otherwise,} \end{cases}$$

Additionally wlog consider $X_i > X_i'$ (the roles of $i$ and $i'$ can be reversed in the following definitions otherwise) and define the auxiliary variables

$$a_i(j) \triangleq \frac{B(r(X_i(j)) + 1) - X_i(j)}{2X^{\max}/(k-1)} \text{ and } b_i(j) \triangleq \frac{X_i'(j) - B(r(X_i'(j)))}{2X^{\max}/(k-1)}$$

$$Z_i(j) = \begin{cases} 0 & \text{w.p. } a_i(j) + b_i(j) \\ 1 & \text{otherwise,} \end{cases}$$

Further define

$$L_{ij} \triangleq |Y_i(j) - Y_i^{\otimes n}(j)| = Z_i(j) \quad \text{if} \quad (i,j) \in S \tag{21}$$

Otherwise if $(i,j) \notin S$ or equivalently $r(X_i(j)) \neq r(X_i'(j))$, we sample the bins independently and the random variables are defined as

$$Y_i(j) = \begin{cases} r(X_i(j)) + 1 & \text{w.p. } \frac{X_i(j) - B(r(X_i(j)))}{B(r(X_i(j)) + 1) - B(r(X_i(j)))} \\ r(X_i(j)) & \text{otherwise.} \end{cases}$$

$$Y_i^{\otimes n}(j) = \begin{cases} r(X_i'(j)) + 1 & \text{w.p. } \frac{X_i'(j) - B(r(X_i'(j)))}{B(r(X_i'(j)) + 1) - B(r(X_i'(j)))} \\ r(X_i'(j)) & \text{otherwise,} \end{cases}$$

Additionally wlog consider $X_i > X_i'$ (the roles of $i$ and $i'$ can be reversed in the following definitions otherwise) and define the auxiliary variables

$$a_i(j) \triangleq \frac{X_i(j) - B(r(X_i(j)))}{2X^{\max}/(k-1)} \text{ and } b_i(j) \triangleq \frac{B(r(X_i'(j)) + 1) - X_i'(j)}{2X^{\max}/(k-1)}$$

$$Z_i(j) = \begin{cases} 0 & \text{w.p. } 1 - a_i(j) - b_i(j) + a_i(j)b_i(j) \\ 1 & \text{w.p. } a_i(j) + b_i(j) - 2a_i(j)b_i(j) \\ 2 & \text{otherwise,} \end{cases}$$

In this case define $L_{i,j} \triangleq r(X_i(j)) - r(X_i'(j)) + 1 + Z_i(j)$ and note that

$$|Y_i(j) - Y_i^{\otimes n}(j)| \leq L_{ij} \tag{22}$$

With these definitions, it can be seen that the marginal distributions of $Y_i(j), Y_i^{\otimes n}(j)$ are equal to the marginal distributions of $U_i^X(j), U_i^{X \otimes n}(j)$ respectively. Further note that since $X_i' = X_i$ for all $i \in [1, n-1]$ we have that $Y_i = Y_i^{\otimes n}$ w.p. 1 for all $i \in [1, n-1]$. Therefore

$$\| \sum_i Y_i - \sum_i Y_i^{\otimes n} \|_q^q = \| Y_n - Y_n^{\otimes n} \|_q^q \leq \sum_j L_{nj}^q,$$

where the inequality follows from (21) and (22). We wish to bound the RHS above. To that end consider the following claim which follows from the definitions.

**Claim 1.**
$$Z_i(j) \leq 2 \quad \text{w.p. 1}$$

$$\mathbb{E}[Z_i(j)] = \begin{cases} a_i(j) + b_i(j) & \text{if } (i,j) \notin S \\ 1 - (a_i(j) + b_i(j)) & \text{otherwise} \end{cases}$$

$$\mathbb{E}[Z_i(j) - \mathbb{E}[Z_i(j)]^2] \leq \begin{cases} a_i(j) + b_i(j) & \text{if } (i,j) \notin S \\ 1 - (a_i(j) + b_i(j)) & \text{otherwise} \end{cases} = \mathbb{E}[Z_i(j)]$$

$$\mathbb{E}[Z_i(j) - \mathbb{E}[Z_i(j)]^4] \leq 4\mathbb{E}[Z_i(j) - \mathbb{E}[Z_i(j)]^2] \leq 4\mathbb{E}[Z_i(j)].$$

Further note that

$$\sum_j \mathbb{E}[Z_n(j)] = \sum_{(n,j)\notin S} (a_i(j) + b_i(j)) + \sum_{(n,j)\in S} 1 - (a_i(j) + b_i(j)) \le \frac{\|X_n - X'_n\|_1}{2X^{\max}/(k-1)}. \quad (23)$$

A direct application of Bernstein's Inequality gives us that with probability at least $1 - \delta/2$

$$\sum_j Z_n(j) \le \mathbb{E}[\sum_j Z_n(j)] + \sqrt{2\mathbb{E}[\sum_j Z_n(j)]\log(2/\delta)} + \frac{4}{3}\log(2/\delta). \quad (24)$$

This gives us that

$$\sum_j |Y_n(j) - Y_n^{\otimes}(j)| \overset{a}{\le} \sum_j L_{nj}$$

$$\overset{b}{\le} \sum_{(i,j)\in S} Z_i(j) + \sum_{(i,j)\notin S} (r(X_i(j)) - r(X'_i(j)) + 1 + Z_i(j))$$

$$\overset{c}{\le} \frac{\|X_n - X'_n\|_1}{2X^{\max}/(k-1)} + \sqrt{2\frac{\|X_n - X'_n\|_1}{2X^{\max}/(k-1)}\log(2/\delta)} + \frac{4}{3}\log(2/\delta)$$

where $a, b$ follow from (21) and (22) and $c$ follows from Claim 1 and (23). This proves the $\ell_1$ norm bound.

We now focus on the $\ell_2$ norm case. For this we note that

$$\forall(i,j) \quad L_{ij} = \begin{cases} \frac{X_i(j)-X'_i(j)}{2X^{\max}/(k-1)} + Z_i(j) - \mathbb{E}[Z_i(j)] & \text{if } X_i(j) \ge X'_i(j) \\ \frac{X'_i(j)-X_i(j)}{2X^{\max}/(k-1)} + Z_i(j) - \mathbb{E}[Z_i(j)] & \text{if } X_i(j) < X'_i(j). \end{cases}$$

Therefore

$$\sqrt{\sum_j L_{nj}^2} = \sqrt{\sum_j \left(\frac{X_i(j) - X'_i(j)}{2X^{\max}/(k-1)}\right)^2} + \sqrt{\sum_j (Z_n(j) - \mathbb{E}Z_n(j))^2}. \quad (25)$$

We now bound $\sqrt{\sum_j (Z_n(j) - \mathbb{E}Z_n(j))^2}$. We can now apply Bernstein's inequality on the random variable $(Z_n(j) - \mathbb{E}Z_n(j))^2$ to get that with probability at least $1 - \delta/2$

$$\sum_j (Z_n(j) - \mathbb{E}Z_n(j))^2 \le \sum_j E[Z_{nj}] + \sqrt{8\sum_j E[Z_{nj}]\log(2/\delta)} + \frac{4}{3}\log(2/\delta), \quad (26)$$

where the RHS uses Claim 1 for bounding expectation and variance.

Therefore combining (25) and (26), we get that

$$\|Y_n - Y'_n\|_2 \le \sqrt{\sum_j L_{nj}^2}$$

$$\le \frac{\|X_n - X'_n\|_2}{2X^{\max}/(k-1)} + \sqrt{\frac{\|X_n - X'_n\|_1}{2X^{\max}/(k-1)} + \sqrt{8\left(\frac{\|X_n - X'_n\|_1}{2X^{\max}/(k-1)}\right)\log(2/\delta)} + \frac{4}{3}\log(2/\delta)}.$$

The proof is finished using a union bound.

$\square$

# E Quantization with Rotation

We prove Theorem 4 here.

**Differential Privacy**
Given any two neighboring data sets $X = \{X_1, \ldots X_n\}, X_{\otimes n} = \{X_1, \ldots X'_n\}$ we define a set of good rotations $U_{\text{good}}$ as follows

$$U_{\text{good}} = \left\{ R \in U | \forall\ i \in [n] \quad \|RX_i\|_\infty \leq \frac{2\sqrt{\log(\frac{2nd}{\delta})}D_2}{\sqrt{d}}, \|RX'_n\|_\infty \leq \frac{2\sqrt{\log(\frac{2nd}{\delta})}D_2}{\sqrt{d}} \right\}$$

where $U$ is the set of $d \times d$ orthonormal matrices. The following lemma follows from [3]. We note that similar analysis holds for uniformly sampled $R$ over real domain and we refer the reader to [10] for details.

**Lemma 8** ([3]).
$$P(HA \in U_{good}) \geq 1 - \delta$$

Let $\text{Rot}(\pi, HA)(X), \text{Rot}(\pi, HA)(X_{\otimes n})$ represent the random output of the protocol $\text{Rot}(\pi, HA)$ on $X, X_{\otimes n}$ respectively and let $S$ be any subset of the output range of $\text{Rot}(\pi, HA)$. Given $\delta$ let $\varepsilon$ be given by Theorem 1 with sensitivity parameters $\{\Delta_1(X^{max}, D), \Delta_2(X^{max}, D), \Delta_\infty(X^{max}, D)\}$. Given a set of vectors $V$ and a rotation matrix $R$ define $R \cdot V = \{Rv | v \in V\}$.

$$Pr(\text{Rot}(\pi, HA)(X) \in S)$$

$$\leq \int_{R \in U_{\text{good}}} (Pr(\text{Rot}(\pi_{sk}(\text{Bin}(m,p)), HA)(X) \in S|R))\, dR + Pr(R \notin U_{\text{good}})$$

$$= \int_{R \in U_{\text{good}}} Pr(\text{Rot}(\pi_{sk}(\text{Bin}(m,p)), HA)(R \cdot X) \in R \cdot S)dR + Pr(R \notin U_{\text{good}})$$

$$\overset{a}{\leq} \int_{R \in U_{\text{good}}} (e^\varepsilon Pr(\pi_{sk}(\text{Bin}(m,p)))(R \cdot X_{\otimes n}) \in R \cdot S) + 2\delta)\, dR + Pr(R \notin U_{\text{good}})$$

$$= \int_{R \in U_{\text{good}}} e^\varepsilon (Pr(\text{Rot}(\pi_{sk}(\text{Bin}(m,p)), HA)(X_{\otimes n}) \in S|R) + 2\delta)\, dR + Pr(R \notin U_{\text{good}})$$

$$\leq e^\varepsilon Pr(\text{Rot}(\pi_{sk}(\text{Bin}(m,p)), HA)(X_{\otimes n}) \in S) + 3\delta$$

$a$ follows from $(\varepsilon, 2\delta)$ differential privacy guarantee for $\pi_{sk}(\text{Bin}(m,p))$ from Theorem 3 and noting that $R \in U_{good}$ in the integral. Hence $\text{Rot}(\pi_{sk}(\text{Bin}(m,p)))$ offers $(\varepsilon, 3\delta)$ differential-privacy.

**Mean Square Error**
The bound on the MSE can be observed by noting that the total change the entire protocol can cause on any individual client vector is bounded by $2D$ in $\ell_2$ norm, therefore the total MSE can be at most $4D^2$ irrespective of the choice of rotation. Therefore

$$\mathcal{E}(\text{Rot}(\pi_{sk}(\text{Bin}(m,p))), HA) = \mathcal{E}(\text{Rot}(\pi_{sk}(\text{Bin}(m,p))), HA|R \in U_{good})+$$
$$\mathcal{E}(\text{Rot}(\pi_{sk}(\text{Bin}(m,p))), HA|R \notin U_{good})$$

$$\overset{a}{\leq} \mathcal{E}(\text{Rot}(\pi_{sk}(\text{Bin}(m,p))), HA|R \in U_{good}) + 4D^2\delta^2$$

$$\overset{b}{\leq} \frac{2\log\frac{2nd}{\delta} \cdot D^2}{n(k-1)^2} + \frac{8\log\frac{2nd}{\delta}}{n} \cdot \frac{mp(1-p)D^2}{(k-1)^2} + 4D^2\delta^2$$

$a$ follows from the argument above and $b$ follows from the MSE guarantee in Theorem 3 and by noting that the rotation is in $U_{good}$.

## Footnotes

[7] we choose $\delta, \delta'$ as $\delta$ in the application of Lemma 6