[Reviews · NeurIPS 2018]

Reviewer 1



--Post rebuttal: The authors' answered my questions satisfactorily, and I raised my score by 1 to 8. I think this is a strong paper. I look forward to reading an updated paper that improves some of the writing and presentation. Summary: This paper seeks to combine two aspects of distributed optimization: (i) quantization to reduce the total communication cost and (ii) privacy through an application of differential privacy. The authors propose to have each of the worker nodes add noise to its gradients before sending the result to the central aggregator node. The challenge in this case is that adding Gaussian noise, as is usual for differential privacy mechanisms, means that the resulting gradients have infinite resolution, so they cannot easily be compressed. Instead, the authors use adding a binomial RV. They analyze this particular differential privacy mechanism. Finally, since the aggregation step requires the central node to estimate the mean, the authors study the error of distributed mean estimation. This is a heavy theory paper. As far as I know, this is the first result of this type, which could be very useful in the federated optimization setting. The results are strong and the theory is a welcome building block for future works. The paper isn't particularly written or easy to understand, with many un-interpreted expressions, but this doesn't take away from the overall contribution. Major comments: - This is a theory paper, so it's important to explain each of the theoretical results. The huge formula in (6) in Theorem 1 isn't very interpretable as it stands. Which one of those three terms is typically dominant? Which of the three sensitivity parameters are typically important for problems of interest? Under what conditions is the resulting epsilon not meaningful (i.e., super huge)? - The role of the scaling term s is also a little bit unclear. It must be sufficiently small (or else the noise magnitudes could be huge, huge enough to be past the representation limit for floats), but if it's too small, the noise may not be able to be distinguishable by floats either. Is there a heuristic rule for getting the right s? - The paper's writing is sometimes pretty rough. Please add some signposting so we know what to expect and in which section. Minor comments: - Line 20: it's not necessarily true that "without a central server, each client saves a global model, broadcasts the gradient to all other clients". It's fine to have each client send its gradient to a potentially small neighborhood of clients (and receive a gradient from each of them). This is the standard setting for decentralized GD and related algorithms, which don't require a complete graph for the topology. There is a whole literature that shows how the graph topology impacts convergence, etc. See Yuan, Lin, and Yin, "On the Convergence of Decentralized Gradient Descent" for a discussion. - What's the reason why you want to use cases other than p=1/2 for the binomial mechanism? - One of the listed improvements is that N >= 8 log (2/delta)/epsilon^2 samples is enough... where does this come out?

Reviewer 2



The authors propose a DP distributed SGD algorithm, which is a composition of randomized quantization and the Binomial mechanism, which can be viewed as a discrete counterpart of the Gaussian mechanism. The paper is clearly written, and all the technical parts seem solid, though I have not fully checked the appendices. A few minor comments: 1) Even though it is hard to analyze "Gaussian noise followed by quantization", it might be helpful if the authors can add experimental results of this scheme, and compare the performance of it with the proposed solution. It might be the case that even though performance analysis is intractable, this simple Gaussian+Quantization scheme might perform pretty well. 2) In Thm. 3, C is claimed to be proportional to "log(k+m) + O(1)". Where does the O(1) term comes from? It seems like that the proposed scheme requires k+m bits per coordinate. 3) Typo: (p6) E((\pi, X^n_1) => E(\pi, X^n_1) == (After rebuttal) I am still not sure why the authors cannot run "1) Add Gaussian noise to discrete input and then 2) quantize it again." Since one can clearly get some DP guarantee for the first step, and the second step does not compromise DP guarantee, one can easily get (eps, delta) of this scheme. This might be a crude upper bound but one can still compare the proposed scheme with this simple baseline. (Actually, this baseline might work pretty well even with their crude privacy guarantees.)

Reviewer 3



The authors study a distributed stochastic gradient in a master-worker setting with two constraints (a) the gradients computed by different worker nodes must be differentially private, and (b) the communication from the worker nodes to the master node should be limited via quantization. the authors design schemes where nodes add noise to their gradients for the sake of differential privacy before sending it to the master for aggregation. The main insight of the paper is that binomial random variables are a good choice for the noise perturbations used by the workers; and that rotational noise can help improve privacy guarantees significantly. The results seem relevant in that it combines two areas of work where there is significant amount of prior work - communication efficiency and differential privacy in distributed SGD. The idea of adding binomial RVs - which are the discrete counter parts of Gaussians - seems natural and intuitive, at least wit hind sight. From a technical viewpoint, it appears that the main contribution is to develop mechanisms for computing the mean of RVs in a differentially private communication efficient manner; since estimating the mean of the gradients reliably leads to reliable estimation of the gradient itself, this leads to reliable estimation of gradients. I found the theorem statements tough to parse and proofs tough to read. I would have liked to see the following comments addressed: --> How does the scheme where noise is added to the gradients prior to quantization perform in comparison with the proposed scheme? --> It seems that the error of the proposed scheme decays as d/n. How does this decay compare with schemes that are not differentially private, or with schemes that are not quantized - a discussion in this regard would help understanding the results better.